# Corrosion-Fatigue Performance of 3D-Printed (L-PBF) AlSi10Mg

**DOI:** 10.3390/ma16175964

**Published:** 2023-08-31

**Authors:** Clara Linder, Flavien Vucko, Taoran Ma, Sebastian Proper, Erik Dartfeldt

**Affiliations:** 1RISE, Corrosion, Vehicle and Surface Protection, Isafjordsgatan 28, 164 40 Kista, Sweden; 2French Corrosion Institute—RISE, 220 rue Pierre Rivoalon, 29200 Brest, France; 3RISE, Manufacturing Processes, Additive Manufacturing, Argongatan 30, 431 53 Mölndal, Sweden; 4RISE, Chemistry and Applied Mechanics, Mechanical Research and Innovation, Gibraltargatan 35, 412 79 Göteborg, Sweden

**Keywords:** atmospheric corrosion, fatigue, additive manufacturing, 3D printing, aluminum alloys, AlSi10Mg

## Abstract

Additive manufacturing (AM) allows for optimized part design, reducing weight compared to conventional manufacturing. However, the microstructure, surface state, distribution, and size of internal defects (e.g., porosities) are very closely related to the AM fabrication process and post-treatment operations. All these parameters can have a strong impact on the corrosion and fatigue performance of the final component. Thus, the fatigue-corrosion behavior of the 3D-printed (L-PBF) AlSi10Mg aluminum alloy has been investigated. The influence of load sequence (sequential vs. combined) was explored using Wöhler diagrams. Surface roughness and defects in AM materials were examined, and surface treatment was applied to improve surface quality. The machined specimens showed the highest fatigue properties regardless of load sequence by improving both the roughness and removing the contour layer containing the highest density of defect. The impact of corrosion was more pronounced for as-printed specimens as slightly deeper pits were formed, which lowered the fatigue-corrosion life. As discussed, the corrosion, fatigue and fatigue-corrosion mechanisms were strongly related to the local microstructure and existing defects in the AM sample.

## 1. Introduction

Lightweight structures and materials are nowadays considered the most effective solutions to decrease energy consumption in the automotive sector [1,2]. Aluminum alloys have been recognized as a viable lightweight alternative to steel because of their good performance and low cost [3]. In the global automotive market, approximately 60% of aluminum components are produced by die-casting. The demand for aluminum die casting is increasing due to both the trend of replacing steel with aluminum and due to the change from forging to casting in vehicle parts [4]. This could potentially lead to new possibilities for additive-manufactured (AM) aluminum components.

AM materials with optimized part design enable further weight reduction compared to conventional manufacturing techniques. The most prevalent AM process techniques for metallic 3D printing are Laser Powder Bed Fusion (L-PBF), Electron Beam Powder Bed Fusion and Direct Energy Deposition [5,6]. The L-PBF process is comparatively robust, productive, cost-effective and provides fine surface finishes [7].

Aluminum alloys with silicon and magnesium as major alloying elements, in particular AlSi10Mg, exhibit moderate mechanical properties, low density and good corrosion performance [8,9]. Mechanical properties obtained by AM processes are similar to cast AlSi10Mg after adapted thermal treatments [10,11,12]. However, the size of Si particles was found to be finer in the case of AM materials. In particular, the morphology and size of eutectic silicon and intermetallic compounds were different, which can greatly impact the mechanical properties of the final products [13]. Fatigue crack initiation is, for example, mainly affected by the presence of porosities or by surface roughness [14,15,16,17,18]. Indeed, two of the main challenges associated with AM materials are the rough surface of as-printed parts and their defect-rich outer layers. Surface treatments, such as mechanical machining or chemical removal by Hirtisation^®,^ can be applied to improve the surfaces and reduce their impact on material properties. The influence of the size and distribution of defects initially present in AM products have been recently investigated for AlSi10Mg alloy [19]. The authors quantified the relationship between defects, fatigue strength and fatigue curves and developed a predictive method that can be applied to perform a defect-tolerance assessment of AM parts. This approach is particularly interesting in predicting the fatigue life of complex components, such as the ones produced by AM.

It is often considered that Al alloys exhibit good corrosion resistance, in particular under atmospheric conditions, due to the presence of a strong passive layer [20]. Nevertheless, they are not immune to localized corrosion, such as pitting, intergranular or exfoliation corrosion, in particular in the presence of chlorides from marine aerosols or de-icing salts used in road environments [21]. The corrosion behavior of cast and AM materials is controlled by surface roughness and inclusions or precipitates [22,23]. Fine Si-rich particles uniformly distributed in the microstructure of AM materials and smoother surface lead to a more uniform passive film and lower sensitivity to localized corrosion compared to cast alloys [24].

Fatigue corrosion has been a concern in the aerospace industry due to the exposure of aircraft components to harsh environmental conditions, notably seawater salts. Several incidents involving fatigue corrosion have been reported in areas such as aircraft structures, landing gear, and engine components [25,26,27]. Fatigue corrosion can also be an issue for automotive applications, particularly in regions where road salts or other corrosive agents are used during winter periods. The combination of cyclic loading during driving and exposure to corrosive environments can lead to fatigue cracks and structural failures in critical components, such as suspension systems, chassis and exhaust systems [28,29]. Under atmospheric corrosion conditions, the fatigue-corrosion cracking mechanism of Al alloys is usually initiated by the formation of corrosion pits or intergranular corrosion, which leads to local acidification and hydrogen production. Then both anodic dissolution and hydrogen embrittlement can take place, depending on the sensitivity of Al alloys and their precipitation state (distribution and composition of intermetallic particles) [30,31,32].

To take corrosion effects into account on the fatigue performance of metallic alloys, pre-corrosion under accelerated corrosion tests and testing in salt solutions or humid air are usually considered [22,31] and have been used for AM Al alloys [33,34,35]. There are other accelerated corrosion tests, such as VDA 233-102 tests (DIN 55635 [36]), that better mimic real outdoor exposure. Combined fatigue and corrosion can lead to rather different results compared to previously mentioned methods [37,38] and is somehow more representative of service life. The quantification of the impact of both mechanisms, i.e., corrosion and fatigue, simultaneously is rather complex compared to a sequential approach. It can be anticipated that running the tests sequentially is less conservative compared to a combined approach. Therefore, the main objective of this study was to investigate the impact of the load sequence, i.e., sequential vs. combined, on the fatigue and corrosion performance of AlSi10Mg specimens. Wöhler diagrams were obtained for nominal (without corrosion), sequential and combined series. Both as-printed and machined surfaces were considered to investigate the impact of surface roughness and defects on fatigue and corrosion. Characterization of the microstructure and initial defects was also carried out to understand better the corrosion, fatigue and fatigue-corrosion mechanisms of AM products.

## 2. Materials and Methods

### 2.1. Printing Process

The AlSi10Mg powder used in this study was provided by SLM Solutions GmbH. The size distribution of the material was 20–63 µm with a mean diameter of 41.7 µm and an apparent density of 1.52 g/cm^3^. The composition of the alloy for the powder was Al (balance), Si (9.9 wt%), Mg (0.34 wt%) and Fe (0.12 wt%). The total amount of other elements was <0.15 wt%.

An SLM 125HL from SLM Solutions GmbH with a YAG fiber laser (Gaussian spot size 70 µm) was used to manufacture the specimens (Figure 1). The printing was conducted in an argon atmosphere to keep the oxygen content below 1000 ppm. The build plate temperature was set to 150 °C. The laser power was set to 350 W, and the laser speed to 1650 mm/s. A strip pattern scan strategy was employed with a 67° rotation between subsequent layers. One border outlining the cross-section and one filled contour between the hatch and the border were used. The hatch distance was set to 0.13 mm, the beam focus was set to 0 mm, and the layer thickness was 0.03 mm.

As-printed specimens were printed using the geometry in Figure 1, perpendicular to the build plate. The specimens were 110 mm high, 24 mm wide and 3 mm thick. The cross-section in the middle of the samples was 20 mm × 10 mm × 3 mm. The machined specimens were printed as rectangular specimens (110 mm × 25 mm × 4 mm) and machined to the same geometry as the as-printed specimens (Figure 1). A total of 96 samples were manufactured on three separate build plates. No heat treatment was applied to the specimens.

### 2.2. Material Characterisation

The surface roughness of specimens was determined (following ISO 4287) using an InfiniteFocusSL microscope and the software Mountain Map on three as-printed and three machined specimens. The analyzed area was 5.7 mm × 5.7 mm in the center of the specimens. The vertical resolution was 0.1 µm, and the lateral resolution was 3 µm.

To investigate defects in the printed materials, cross-sections were taken in the middle of three specimens, polished and characterized using a Leica stereomicroscope. Fractured surfaces were observed with a JEOL JSM-6610 scanning electron microscope (SEM) with an acceleration voltage of 10 kV. Cross-sections of a corrosion pit were observed with a Sigma 300 VP Gemini SEM (Zeiss, acceleration voltage of 10 kV) with an energy dispersive spectroscopy (EDS) detector (Oxford Instruments X-MAx^N^, Stockholm, Sweden) integrated into the microscope.

### 2.3. Atmospheric Corrosion Testing

Specimens were exposed in a ControlArt Test chamber Type 2 following the standard corrosion cycle VDA 233-102 [36] (see Appendix A). Three different cycles with varied temperature (−15 °C to 50 °C), relative humidity (50 to 100%) salt spray (1% NaCl, pH 6.5–7.1) were used. The total exposure time was 6 weeks (42 days). Prior to exposure, specimens were cleaned with ethanol and masked in the grip areas. Specimens were placed at a 20° inclination angle from the horizontal plane. After exposure, specimens were rinsed with deionized water and dried with hot air.

Corrosion products were removed by pickling with nitric acid (68%, room temperature). Three as-printed and three machined specimens were observed with an InfiniteFocusSL microscope. The software Mountain Map was used to determine the average pit depth from the 10 deepest pits on the samples based on the height profile of the specimens. The pit density was determined from a 25 mm^2^ area with pits that were at least 5 µm deep for the machined specimens and 15 µm for the as-printed specimens.

### 2.4. Fatigue Testing

The nominal (no corrosion) and sequential (after corrosion) fatigue testing was carried out with an Instron 1341 machine equipped with a ±100 kN load cell (valid calibration class 0.5). All tests were carried out at a frequency f = 10 Hz and a load ratio R = 0.1. Run-out was set to N = 2 × 10^6^ cycles.

The combined fatigue and corrosion testing was carried out using in-house developed equipment [36]. Pneumatic actuators with a load capacity of up to 15 kN were individually controlled to apply fatigue cycles at f = 0.5 Hz and R = 0.1. The whole structure was placed inside a ControlArt Test chamber Type 2 to apply the VDA 233-102 cycle. Specimens were placed at a 20° inclination angle from the horizontal plane. The maximum exposure was 46 days (2 × 10^6^ fatigue cycles).

## 3. Results and Discussion

### 3.1. Material Characterisation

The surface roughness of as-printed and machined specimens was measured prior to fatigue and corrosion testing. The average surface roughness (R_a_) values were 4.0 ± 0.6 µm and 0.5 ± 0.1 µm for as-printed and machined specimens, respectively.

Cross-sections of untested specimens were analyzed with a stereomicroscope to highlight porosities as white circles in Figure 2a,b. SEM micrographs were also taken in the bulk of the alloy parallel to the building direction to identify the microstructure (Figure 2c).

In Figure 2a, large amounts of porosity were observed in the contour and filled contour area for as-printed specimens. The contour is part of the printing strategy in the L-PBF process. For machined specimens, Figure 2b, there were considerably fewer pores in the material as the contour has been removed. 

Figure 2c shows the typical microstructure of an L-PBF AlSi10Mg alloy with its columnar Al phase (dark contrast matrix) surrounded by eutectic Si particles (white particles network). Compared to cast materials, the AM microstructure is more homogenous; the Si particles are evenly distributed over the Al matrix and are much smaller than in cast materials [33].

### 3.2. Corrosion Testing

After 6 weeks of VDA, 233-102 corroded specimens from the sequential testing were analyzed. Figure 3 shows photographs of as-printed and machined specimens after the corrosion test.

For both as-printed and machined specimens, the corrosion was uniformly spread over the exposed surface. On the as-printed specimens, there were spot-like products on the surface (Figure 3a). For the machined specimens, the corrosion products were different from the as-printed specimens, e.g., the color of the specimens was different (Figure 3b). To further characterize the extent of the corrosion, three as-printed and three machined specimens were pickled with nitric acid to remove the corrosion products, leaving only the bare metal, which allows a more accurate pit depth analysis. A summary of the analysis is shown in Table 1.

The as-printed specimens had the deepest pits and highest average pit depth of 50 µm, almost five times higher than the average pit depth for the machined specimens. The removal of surface defects and porosities, combined with the reduction of surface roughness, which decreased the sensitivity to localized corrosion, had a positive effect on the corrosion pit initiation and propagation. A similar effect has been reported for AlSi10Mg after polishing [34] and shot-peening [23].

The spots seen in Figure 3a were revealed to be corrosion pits. They were found over the whole surface of the exposed area but not at the same locations from specimen to specimen. A cross-section of a corroded as-printed specimen is shown in Figure 4.

The Si network was the only material remaining at the bottom of the pits, as seen in the EDS maps (Figure 4). This indicates that the Al matrix has preferentially been dissolved. This is caused by the potential differences between Al and Si, which leads to a galvanic coupling and, consequently, a preferential dissolution of Al [24]. Because of the more even distribution of Si particles in the Al matrix (Figure 2c), the AM alloys become less sensitive to localized corrosion compared to cast alloys at similar compositions [33,39].

### 3.3. Fatigue

Results from the fatigue testing are shown in Figure 5, with data points from machined and as-print specimens plotted using different symbols. In addition, bounds have been included for the two series to show that the fatigue performance of machined specimens is consistently better than that of the as-print specimens. It is, therefore, concluded that removing the outer surface layer and reducing the surface roughness (from 4 to 0.5 µm) by machining significantly improved the fatigue performance. This effect has also been shown for other AM Al alloys [14,15].

Results for the machined specimens are shown in Figure 6 using different markers for the different test modes. Bounds are included for each individual series. Even though the number of data points is relatively low, it is observed that the fatigue limit (i.e., the stress level for which N > N_VDA_) seems to be lowered when changing from the nominal mode to the sequential mode. This effect is even more pronounced for the combined testing. It is also observed that the results from the combined testing seem to converge to that of the nominal testing for decreasing cycles to failure. This is explained by the fact that lower cycle numbers also mean that the time that the specimens are exposed to the corrosive environment decreases. In contrast, the fatigue response from the sequential series should be consistently lower than the nominal series since these specimens have been exposed to a full VDA cycle prior to the fatigue testing. Interestingly, this means that there should be a transition point at which the combined approach becomes more conservative than the sequential approach. This transition seems to occur at around ∆σ ≈ 150 MPa, although more testing is required to determine this more rigorously. Nonetheless, the fatigue and fatigue-corrosion performance of the machined Al alloy in this study is comparable to or higher compared to other reports [14,15,40,41].

Results for the as-print specimens are compared in Figure 7. As shown, the fatigue performance in air (nominal) was better than after corrosion exposure using sequential or combined modes. It should be pointed out that the fatigue limit was particularly low in all cases (down to 45–63 MPa), so the difference was less clear between the different fatigue modes. However, despite the limited number of tests, the drop in fatigue performance was even higher under combined fatigue-corrosion conditions compared to sequential fatigue and corrosion, without a marked transition as for results on machined specimens.

For the sequential and combined fatigue-corrosion testing, the range of low-stress amplitudes was mainly explored. Indeed, for high-stress amplitudes, the number of cycles to failure is decreased so that the time under the corrosive environment is short. Under such conditions, no significant effect from the corrosive environment is expected, particularly in the case of aluminum alloys with slow corrosion kinetics.

To summarize the results from the fatigue testing, it is concluded that the combined testing is a more conservative approach than the sequential approach for cycle numbers beyond the transition point. The VDA 233-102 selected in this study has been largely used in Europe and, in particular, by the German automotive industry. However, these test conditions might not be severe enough for testing the Al alloy in this study as the corrosion pits were relatively narrow (<200 µm [41]). The effect of corrosion on fatigue could have been more prominent if a different environment (e.g., AlCl_3_ salts [42]) or longer exposure times (>2 × 10^6^ cycles) had been chosen. The selection of the right testing environment but also testing sequence (sequential or combined) plays an important role in the assessment of AM Al alloys as the printing and post-processing techniques (e.g., grinding, Hirtisation^®^) become more mature.

### 3.4. Fractography

To further investigate the differences in fatigue performance for the different load sequences, fractography with SEM was carried out. Figure 8 shows a fractography SEM micrograph of as-printed specimens after the different load sequences.

For all load sequences, the crack initiated at one of the corners of the specimens. For the sequential and combined sequences, the location of the crack was slightly displaced from the corner and towards the top surface due to pit formation close to this area of high stress. Indeed, the corners of the AM specimens were particularly sensitive to stress as it was the intersection between two print layers where the cross-sectional area was smaller, which caused an increase in stress levels at that location. Corners also had porosity clusters (Figure 2a); thus, more defects are present that can initiate a crack [40,43]. Furthermore, layer interfaces and partially melted powders in as-printed specimens were weak regions and likely to be where the fatigue cracks initiate [43]. 

In the combined load sequence, secondary cracks could be observed at the pit locations (Figure 8d). As pit depth and pit growth rate are controlling the conditions for nucleation of fatigue-corrosion crack [44], this indicates that even though the corrosion conditions were mild and the pits narrow (<200 µm), fatigue-corrosion was promoted. Local fatigue crack growth rate and cracking mechanism for materials subjected to fatigue-corrosion can be investigated through striations patterns on fractured surfaces [45]. In this study, however, such patterns were not visible at the crack initiation location. As the microstructure of the Al alloy has refined the Si network (Figure 2c), such striations might not have been visible at the scale of the microstructure. It could also be because of the corrosion and dissolution along the crack path, which could have erased the patterns.

Figure 9 shows the fractography of a machined specimen subjected to nominal and combined load sequences.

The crack initiation in corrosion pits was not observed for machined specimens subjected to sequential and combined loads. The crack was found in the corner of the specimen in the same manner as the nominal condition. This means the corrosion did not have an impact on the crack initiation mechanism, which was expected given the limited pit depth (11 µm). 

Overall, the low pit depth on as-printed and machined specimens made the fatigue-corrosion mechanism more influenced by the surface state and geometry of specimens. In all specimens, the main fatigue crack started systematically from the corner of the specimens, even if secondary cracks could be observed from pits (Figure 8d). The rectangular section of the specimen might have minimized the impact of pitting on the surface and enhanced the impact of pits located close to the corners.

For the characterization of the corrosion under accelerated corrosion tests, flat corrosion coupons offer better control of corrosion degradation. The side facing the salt spray in the corrosion chamber acts as a collector of the salt deposition and will be particularly degraded. Thus, flat tensile specimens were selected for fatigue-corrosion testing to benefit from the knowledge of accelerated corrosion tests. However, this geometry leads to over-stress distribution at the sharp edges of the specimens, promoting crack initiation at this location. The use of round tensile specimens would solve this mechanical problem but raise others. Indeed, under atmospheric corrosion tests, the accumulation of contaminants on small cylinders is likely not favored, and some saltwater droplets can be formed below the cylinder, drastically modifying the local corrosion conditions. Optimization of the geometry of the fatigue-corrosion specimens might be required for future studies.

## 4. Conclusions

The corrosion and fatigue performance of an L-PBF AlSi10Mg alloy has been investigated based on Wöhler diagrams. Atmospheric corrosion testing was conducted according to the standard accelerated corrosion test VDA 233-102. The impact of load sequence, i.e., sequential or combined fatigue and corrosion loading, was compared. The influence of the surface condition (as-printed or machined) on the results was also studied. Additional characterizations of the fracture surface after fatigue testing were conducted. From the results, the main conclusions of this study are the following:Machining specimens removed defect-rich layers such as porosity and partially melted powder and reduced the surface roughness. Machined specimens had considerably higher fatigue performance than as-printed specimens, regardless of loading sequence;The pit depth was lowest for machined specimens; thus, the effect of corrosion on fatigue was not as significant as for as-printed specimens;Differences between the loading sequences were evident, i.e., the fatigue performance was reduced when changing from nominal to sequential to combined sequence for both as-printed and machined specimens;All specimens had their main fatigue crack initiated at the corners of the specimens, where porosities and stress from AM process were concentrated;For sequential and combined testing, the crack was initiated in corrosion pits close to the corners of specimens;Evidence of a fatigue-corrosion mechanism was found for the as-printed specimens, even though the corrosion pits were narrow.

## Figures and Tables

**Figure 1 materials-16-05964-f001:**
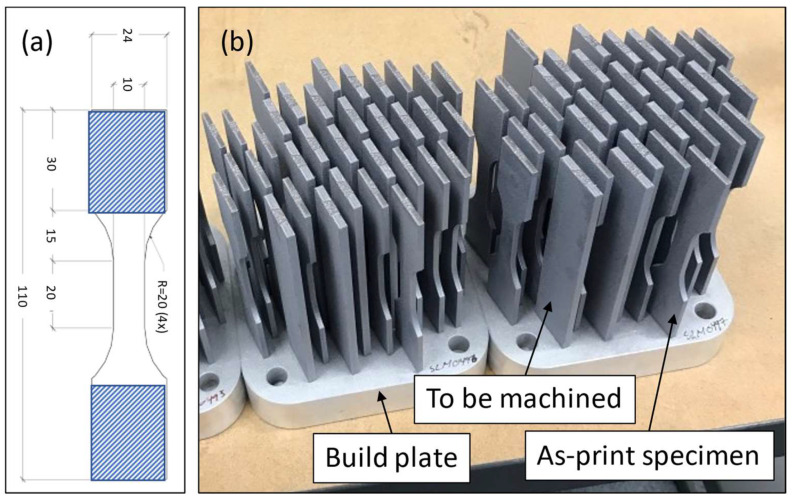
(**a**) Test specimen geometry (the gripped area is highlighted)—dimensions in millimeters. (**b**) Printed specimens before being removed from the build plate.

**Figure 2 materials-16-05964-f002:**
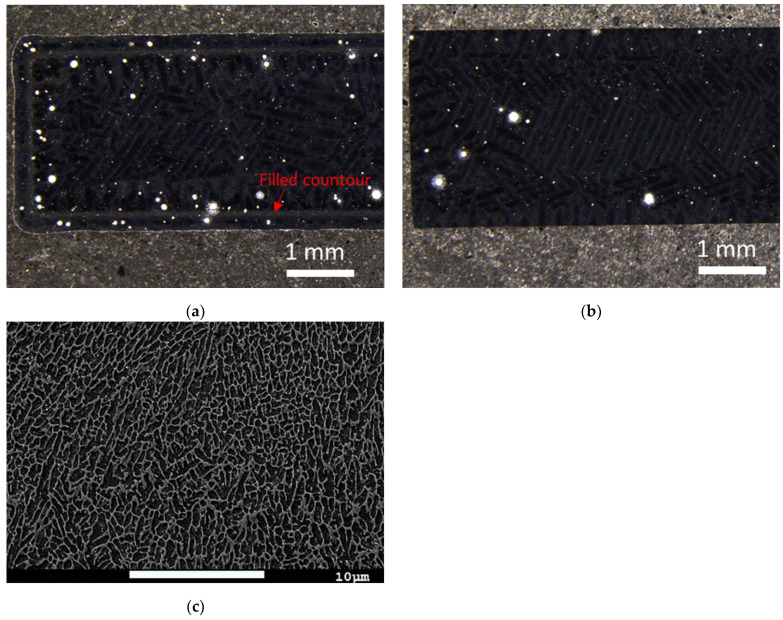
Stereo optical micrographs of cross-sections highlighting the porosities as white circles for (**a**) as-printed specimens, the arrow indicates the filled contour from the printing process; (**b**) machined specimens; (**c**) SEM micrograph (×2500, 5.0 kV) of a cross-section parallel to the building direction, white particles are the Si-rich network, dark contrast is the Al matrix.

**Figure 3 materials-16-05964-f003:**
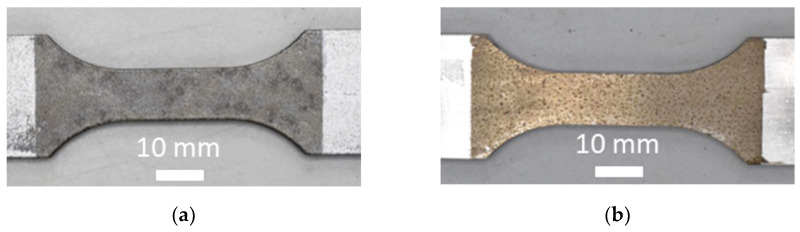
Photograph of corroded specimens from sequential testing after 6 weeks of VDA 233-102 accelerated corrosion cycle (**a**) as-printed and (**b**) machined.

**Figure 4 materials-16-05964-f004:**
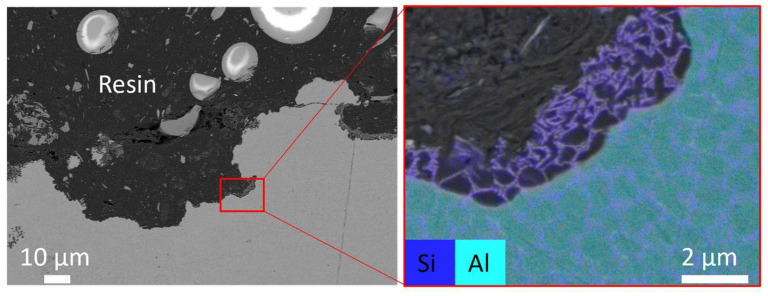
SEM micrograph of the cross-section of the corrosion pit in an as-printed specimen and EDS maps.

**Figure 5 materials-16-05964-f005:**
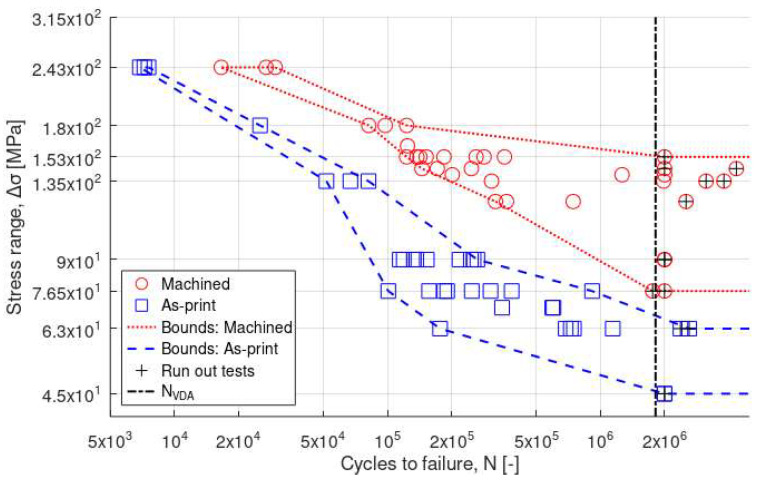
Stress range vs. cycles to failure (log–log) for all tests. Bounds have been added to illustrate the separation between the machined specimens (red circles) and the as-printed specimens (blue squares). The vertical dotted line corresponds to the number of cycles in a full VDA cycle (N_VDA_ ≈ 1.8 × 10^6^ cycles). Run-out tests are indicated by a +.

**Figure 6 materials-16-05964-f006:**
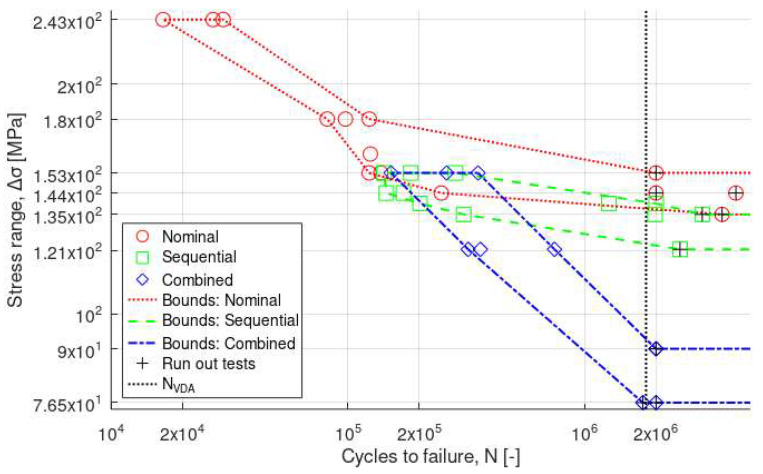
Stress range vs. cycles to failure (log–log) for the machined specimens.

**Figure 7 materials-16-05964-f007:**
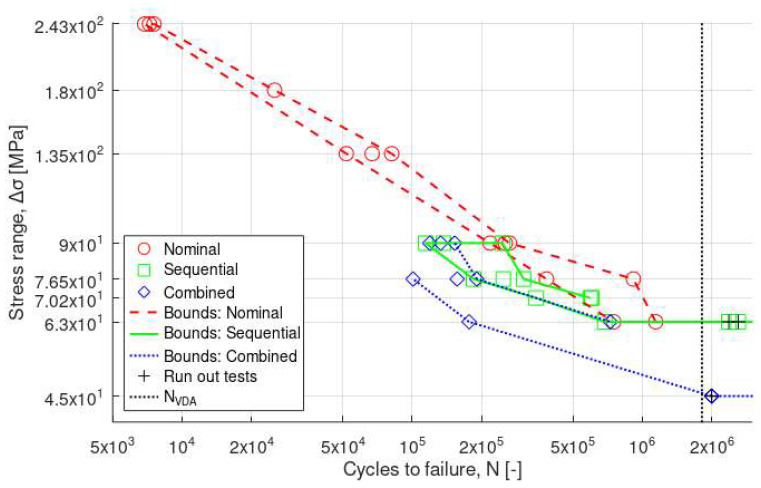
Stress range vs. cycles to failure (log–log) for as-print specimens.

**Figure 8 materials-16-05964-f008:**
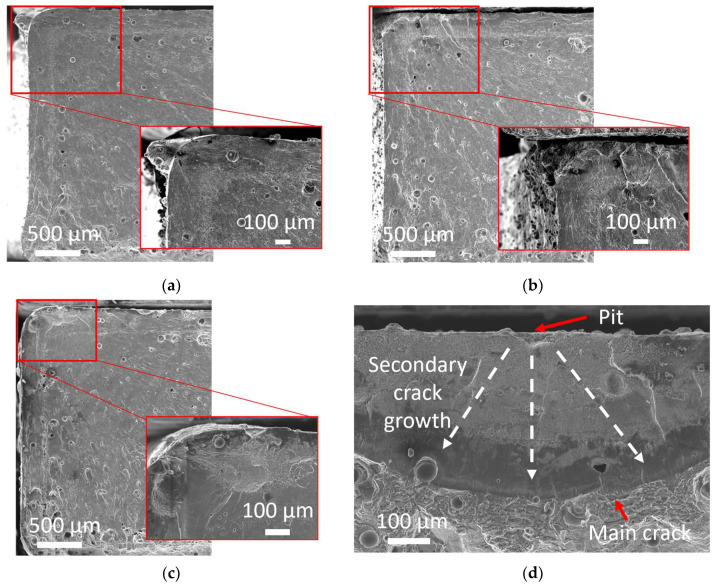
Fractography SEM micrographs of fatigue cracked as-printed specimens subjected to load sequence (**a**) nominal; (**b**) sequential; (**c**) combined; (**d**) secondary crack growth after combined load sequence N = 119,289 cycles, σ_max_= 100 MPa.

**Figure 9 materials-16-05964-f009:**
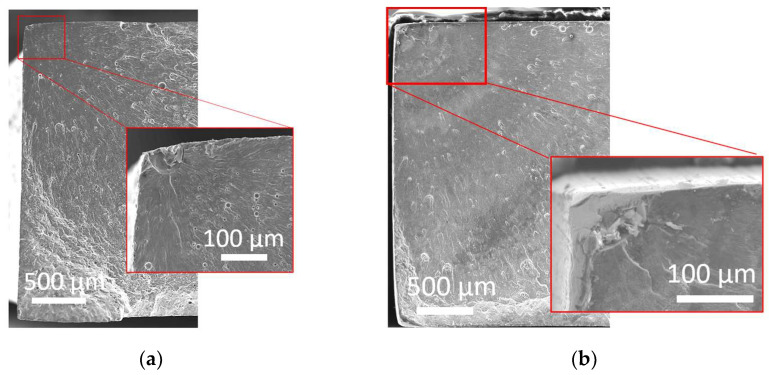
Fractography SEM micrographs of fatigue cracked machined specimens after load sequence (**a**) nominal; (**b**) combined load sequence N = 119,289 cycles, σ_max_= 100 MPa.

**Table 1 materials-16-05964-t001:** Average and maximum pit depth for as-printed and machined specimens after 6 weeks of VDA 233-102.

Surface Condition	Average Pit Depth [µm]	Maximum Pit Depth [µm]	Pit Density [pit/cm^2^]
As-printed	50 ± 12	65	536 ± 20
Machined	11 ± 1	14	140 ± 4

## Data Availability

The data presented in this study are available on request from the corresponding author.

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
