# Peer review of "Corrosion-Fatigue Performance of 3D-Printed (L-PBF) AlSi10Mg"

_materials, 2023, doi:10.3390/ma16175964_

Round 1

Reviewer 1 Report

This study investigates the corrosion-fatigue performance of 3D printed Al alloy by subjecting specimens to an atmospheric corrosion cycle and mechanical fatigue loading using three different load sequences: without corrosion, after the corrosion cycle, and during the corrosion cycle. The title of the article is beautiful for readers and practical. The article is well organized, but the following points should be considered.

The abstract must be completely changed and rewritten. Innovation, conducted tests, achievements, and quantitative results are not presented. The purpose of research and innovation should be clearly stated. Also, the performed tests should be presented first, and then the results should be presented quantitatively and qualitatively.

The abstract should be more attractive. The current version presented predictable results, for example, the effect of surface roughness on fatigue behavior.

The text of the article needs basic writing and grammar corrections.

Referencing the articles is disappointing (Lines 44, 48, 56, and 74). The use of general sentences with more than four references can be seen in the first paragraph of the introduction. On the other hand, appropriate references were not used to analyze the results.

The introduction is very general. Although the introduction is long, it is written superficially in some paragraphs. Also, in the end, a suitable summary of the importance of the present issue should be provided.

Use the following resources to deepen the introduction. Defect tolerant fatigue assessment of AM materials: Size effect and probabilistic prospects. Probabilistic fatigue modelling of metallic materials under notch and size effect using the weakest link theory. Jointing of CFRP/5083 Aluminum Alloy by Induction Brazing: Processing, Connecting Mechanism, and Fatigue Performance.

It is suggested to summarize the physical and mechanical properties of the raw material in a table. In a similar way, printing parameters can also be presented in a table. How is the printing quality confirmed and checked? How has the reproducibility of these results been checked?

Lines 246-251 are dumb; please clarify. Is the stress level considered the same for all samples? Add the error bar to the results. In the conclusion section, a summary of the purpose of the research, innovation, and research method should be presented before presenting the highlights.

Can printing parameters directly affect the final part's fatigue behavior?

It is suggested that, if possible, the porosity (density) of the printed samples should be provided as a quantitative measure of printability. Or to ensure that the printed samples have the highest print quality according to the previous sources.

no comment.

Reviewer 2 Report

I have no comments, but there are two wishes to the authors:

1) Define the notation Ra (line 160) and explain how Ra is determined with an accuracy of 0.001 µm if "the vertical resolution was 0.1 µm".

2) Explain why two cyclic loading frequencies were used. Is there any certainty that the results of the nominal and sequential testing do not depend on the frequency?
